# Neoplasms in Domestic Ruminants and Swine: A Systematic Literature Review

**DOI:** 10.3390/vetsci10020163

**Published:** 2023-02-18

**Authors:** Jackson Vasconcelos, Maria dos Anjos Pires, Anabela Alves, Madalena Vieira-Pinto, Cristina Saraiva, Luís Cardoso

**Affiliations:** 1CECAV—Animal and Veterinary Research Centre, Associate Laboratory for Animal and Veterinary Sciences (AL4AnimalS), University of Trás-os-Montes e Alto Douro (UTAD), 5000-801 Vila Real, Portugal; 2Department of Veterinary Sciences, University of Trás-os-Montes and Alto Douro (UTAD), 5000-801 Vila Real, Portugal

**Keywords:** bovine, domestic ruminants, goats, neoplasms, risk factors, sheep, swine, systematic review

## Abstract

**Simple Summary:**

Given the scarcity of information and the lack of comprehensive studies on neoplasms in domestic ruminants, i.e., cattle, sheep, and goats, and domestic pigs, the aim of the present study was to systematically review the scientific literature to verify the occurrence, type, organ system, and organs most affected by neoplasms in domestic cattle, sheep, goats, and pigs. Based on the results, the bovine species was the most affected by the neoplasm and also the most studied in relation to small ruminants and pigs. In all species, the most affected organ system was the integumentary system, and the most frequent neoplasms were squamous cell carcinomas for cattle, goats, and sheep, while melanoma was the most frequent for pigs. Few studies were carried out in slaughterhouses, and those found to be in the majority referred to cattle and pigs, none of them mentioned goats and sheep. No studies were found that measured the economic losses associated with the condemnation of carcasses of the studied species. The results reinforce the need to carry out studies on farms and slaughterhouses, which would provide more information such as the total number of animals and the origin of the samples.

**Abstract:**

Background: Due to the limited information and lack of studies on neoplasms in domestic ruminants, i.e., cattle, sheep, and goats, and domestic swine, the objective of the present study was to systematically review the scientific literature to verify the occurrence, type, organ system, and organs most affected by neoplasms in these animals. Methods: The recommendations of the PRISMA methodology were followed for the elaboration of this study. The research consisted of a systematic review of neoplasms in domestic cattle, sheep, goats, and swine. Results: The number of neoplasms found was 1873. The most affected organ system was the integumentary system with 35.0%, followed in descending order by the alimentary system with 16.90%, the hematopoietic system with 13.50%, the special senses (i.e., eyes and ears) with 10.51%, the female and male genital systems with 7.31%, the urinary system with 4.38%, the liver and biliary system with 3.152%, the endocrine glands with 3.91%, the respiratory system with 2.67%, the nervous system with 2.35%, bones and joints with 0.43%, muscles and tendons with 0.37%, the cardiovascular system with 0.21%, and the pancreas with 0.16%. Of the animals with neoplasms studied, cattle were affected in 69.80% of cases, goats in 10.52%, sheep in 10.46%, and swine in 9.18%. In all species, the most frequent neoplasms were squamous cell carcinomas in ruminants, while melanoma was the most frequent in swine. Few studies carried out in slaughterhouses were found, and the existing ones referred to cattle and swine. No data were found on economic losses with carcass condemnation. Conclusions: In view of the above, it is necessary to carry out extensive and detailed studies that provide knowledge about the impact of neoplasms on the production and condemnation of carcasses in domestic cattle, sheep, goats, and swine and the respective risk factors.

## 1. Introduction

Increased meat production is required due to rising global population and demand for animal protein in food. According to the Food and Agriculture Organization of the United Nations [1], in the last year the production of meat from ruminants and pigs in the world reached 352.7 million tons. Given the importance of these species as a source of animal protein and the increase in their consumption, associated with the rise in herds and production, the diagnosis of neoplasms has increased [2].

Slaughterhouses for ruminants and swine are important sources of disease detection in food-producing animals, and through retrospective and prospective studies in these places, pathologists contribute to the knowledge of a specific disease or diseases of an entire organ system [3,4].

Among the lesions detected in the post-mortem examination and, eventually, in the ante-mortem are neoplasms [5]. These may have their origin in inherited genetic alterations that will be present in all cells within the organism or in somatic alterations that accumulate in individual cells and tissues of the body over time and may have their origin in chemical, physical (radiation), or infectious causes [6]. As an example of an infectious cause, lymphosarcomas, which are the most frequently found neoplasms in data collection from laboratories [7] and slaughterhouses [4], are second only to squamous cell carcinoma (SCC) [8].

Given the limited information and the lack of comprehensive studies on neoplasms in ruminants and swine, the objective of the present study was to systematically review the scientific literature to verify the occurrence, type, organ system, and organs most affected by neoplasms in ruminants and swine. The review was carried out through the selection of articles, case reports, retrospective works, and experimental reproductions published in English, Portuguese, and Spanish in Lilacs, PubMed, SciELO, Science Direct/Elvesier, and Google Scholar databases.

## 2. Materials and Methods

The recommendations of the PRISMA (Preferred Reporting Items for Systematic Reviews and Meta-Analyses) methodology [9] were followed for the preparation of this study. The research consisted of a systematic review of neoplasms in ruminants and swine.

### 2.1. Inclusion and Exclusion Criteria

As inclusion criteria, full articles and short communications published between 2002 and 2021 in indexed journals with relevant information related to the topic of neoplasms in ruminants and swine were considered eligible. As exclusion criteria, conference proceedings, book chapters, and technical manuals were not considered. Works that were inaccessible were also discarded. In view of these criteria, papers were read in full, and after reading, studies lacking relevant data were excluded.

### 2.2. Sources of Information and Search Strategy

Articles and case reports were consulted in the electronic databases PubMed (National Library of Medicine, National Institutes of Health, Bethesda, MD, USA), SciELO (Scientific Electronic Library Online), Science Direct/Elsevier, and Google Scholar, published in English, Portuguese, or Spanish between 1 January 2002 and 31 December 2021. In the search, the descriptors used were “neoplasm OR tumor* OR tumors” and “bovine OR caprine* OR cattle* OR goat* OR ovine OR pig OR pigs OR ruminant OR sheep OR swine”. Filters were used to locate the words only described in the title of the article in English, Portuguese, or Spanish.

### 2.3. Selection of Studies and Data Extraction

Two researchers independently performed a selection of studies by analyzing the title and abstract and then reading the full text. Complete evaluations of articles and case reports were performed. There were no divergent cases between the two researchers.

Information considered relevant and important for the review was collected and described, such as the type of neoplasm, in which organ/system it is located, the country where the work was performed, and the sample origin (slaughterhouse, laboratory, or farm).

### 2.4. Data Analysis

Relevant information and results from eligible articles included in this review were analyzed and described using a descriptive analysis.

### 2.5. Characterization and Classification of Ruminant and Swine Neoplasms

The neoplasms of ruminants and swine described in the studies eligible for this review were entered according to the organ system affected, as described by Maxie and Miller [3]. The classification used for the neoplasms described in the included studies followed the classification of the World Health Organization (WHO) for domestic animals [5].

## 3. Results

Through this search procedure, 392 studies were identified that addressed neoplasms in ruminants and swine. The 392 studies were obtained from the following electronic databases: PubMed = 128; SciELO = 52; Science Direct/Elsevier =183; and Google Scholar = 29. Inclusion criteria led to the selection of 70 publications that were read in full (Figure 1).

In the evaluation of the studies, the following aspects were considered: neoplasms in cattle, sheep, goats, and swine, diagnosed from slaughterhouses or farms; retrospective studies carried out in pathological anatomy laboratories or in official databases; or case reports, which contained the following elements: the occurrence and classification of each type of neoplasm by species; the organs most affected; and the main diagnostic methods in ruminants and swine.

As for the studies surveyed, 36 were full papers and 34 were case reports. Regarding the number of studies per species, 60 studies reported neoplasms in ruminants (25 in cattle, 18 in goats, and 16 in sheep) and 20 in swine. When the articles and case reports were added up, a total of 79 studies were obtained, as some reported this set of conditions in more than one species. Considering the origin of cases of neoplasms in ruminants and pigs, only 35.7% (25/70) of studies bring this information to light. Regarding sampling, only 17.0% (12/70) of studies contained this type of information. Of the 12 studies with sampling, five were in cattle, three in goats, four in sheep, two in pigs, and two in goats and sheep together. Of the 1873 neoplasms reviewed, 66.6% (1248/1873) are epithelial, 32.1% (602/1873) mesenchymal, and 1.2% (23/1873) are embryonic neoplasms.

A summary of the distribution of the number of studies, correlated with the country where they were carried out and the species studied, can be seen in Table 1. Most of the studies reviewed did not report the origin of the cases of neoplasms according to species (Table 2). From the 70 studies, it was possible to extract a total of 1873 cases of spontaneous neoplasms reported in ruminants and swine. Of the neoplasms identified in this review, 1308 were reported in cattle, 197 in goats, 196 in sheep, and 172 in swine. For the classification into epithelial, mesenchymal, and embryonic neoplasms, the results are described in Table 3. The distribution of neoplasm cases by species and organ system can be seen in Table 4, Table 5, Table 6 and Table 7.

Based on the searched literature, 1873 neoplasms were found. The organ system that was described with the highest number of neoplasms was the integumentary system with 35.0% (655/1873), then in decreasing order were: the alimentary system with 16.90% (316/1873), the hematopoietic system with 13.50% (253/1873), the special senses sensory system with 10.51% (197/1873), the male and female genital systems with 7.31% (137/1873), the urinary system with 4.384% (80/1873), the hepatic and biliary systems with 3.152% (59/1873), endocrine glands with 3.091% (58/1873), the respiratory system with 2. 67% (50/1873), the nervous system with 2.354% (44/1873), bones and joints with 0.43% (8/1873), muscles and tendons with 0.374% (7/1873), the cardiovascular system with 0.21% (4/1873), and the pancreas with 0.16% (3/1873). Of the 1873 neoplasms surveyed, 1308 were diagnosed in cattle, which represents 69.80% (1308/1873) of the cases, goats with 10.52% (197/1873), sheep with 10.46% (196/1873), and pigs with 9.18% (172/1873). As observed in Table 2, the bovine species was the most studied, with the highest number of articles and case reports found in the researched databases and the one most affected by neoplasms in the 70 studies reviewed. The results extracted from the reviewed studies indicate that the organ/system most affected by neoplasms in cattle was the integumentary, with 28.80% (376/1308) of the neoplasms diagnosed in this species. Papillomas/fibropapillomas were the most frequent neoplasms in the integumentary system of cattle, with 54.5% (216/376) of cases, followed by squamous cell carcinomas with 25.8% (97/376). The other neoplasms, as well as the other organ systems affected by neoplasms, are detailed in Table 5.

Among the reviewed studies, 18 contained information on neoplasms in goats, which totaled 197 neoplasms, representing 10.5% (197/1873) and being the second most affected species. In this species, the most frequently diagnosed neoplasm was SCC (61/197, 31.0%). However, the most affected organ system was the integumentary (95/197, 48.2%). The distribution of the other types of neoplasms and the organ systems affected are described in Table 6.

Sixteen studies on neoplasms in sheep were included in this review, and 196 neoplasms were described. The most commonly diagnosed neoplasm in this species was SCC (119/196, 60.7%), and the tegumentary system was the most affected (116/196, 59.2%). The other results for other types of neoplasms and the affected organ systems are shown in Table 7.

Twenty studies on neoplasms in swine were included in the review. Of the 20 studies reviewed, 172 neoplasms were identified, representing 9.18%. Of the neoplasms found in swine, melanoma was the most diagnosed (34/172, 18.9%), and the integumentary system was the most affected (68/172, 39.5%). The other neoplasms found, along with the affected organ systems, are presented in Table 7.

## 4. Discussion

Neoplasms in production animals increase economic losses, but surveys on their occurrence are scarce [5,10]. According to the articles and reports reviewed for this review, in the first 20 years of this century, neoplasms in domestic ruminants [7,8,11,12,13,14] and in pigs [15] were studied and diagnosed with greater frequency worldwide. From the 70 studies reviewed, it was possible to extract a total of 1873 cases of spontaneous neoplasms reported in ruminants and swine. Of the neoplasms identified in this review, 1308 were reported in cattle, 197 in goats, 196 in sheep, and 170 in pigs. Among domestic ruminants and pigs, the most affected organ system was the integumentary (cattle = 28.74% (376/1308), goats = 47.74% (95/199), sheep = 59.20% (116/196); swine = 40.00% (68/170). The most diagnosed neoplasm was SCC in ruminates and melanoma in swine.

### 4.1. Cattle

The bovine species presented the highest number of neoplasms in the studies reviewed [2,7,16]. This species has the highest number of diagnoses of neoplasia, probably due to the large herds distributed around the world associated with increased production in other countries and the greater consumption of beef in the world [1].

The organ system most affected, according to the studies reviewed, by neoplasms in cattle was the integumentary system; the same was observed in a previous study carried out with neoplasms in production animals [7]. However, the most commonly diagnosed neoplasms were papillomas/fibropapillomas, with 54.45% (216/376) of the cases. This trendfor papillomas was also observed in a study conducted in Egypt [16], but they did not identify which of the papillomaviruses were involved. In the literature, 13 types of bovine papillomavirus (BPV 1-13) are described for IS; the types involved are (BPV–1; BPV–2; BPV–3; BPV–5; BPV–8; and BPV–11) [17]. Bovine papillomatosis (papillomas/fibropapillomas) is a herd problem as it is easily transmitted by animal-animal contact and by fomite [18,19]. Most affected animals recover on their own, but in some cases, the papillomas can persist for up to 6 months or longer, resulting in loss of production and weight loss [20]. 

Squamous cell carcinoma was the second most commonly diagnosed neoplasm in IS, with 19.76% (97/491). Similar results were previously described by several authors [8,13,16,21,22]. It is also important to note that for the bovine species, independently of the organ system, SCC was the most commonly diagnosed neoplasm [4,7,13,14,16,21].

The second organ system with the highest number of neoplasms diagnosed according to the current review was the alimentary system, with 18.80% (246/1308). Squamous cell carcinoma with 81.82% (216/264) and papilloma/fibropapilloma with 11.00% (29/264) were the most frequent neoplasms, respectively. Similar results were found in a survey on the frequency of tumors in cattle from samples from slaughterhouses [7,23,24].

According to the studies reviewed, the hematopoietic system was the third most affected organ system, with 15.14% (1988/1308) of diagnosed neoplasms, and lymphoma and/or lymphosarcoma were the most common neoplasms in cattle. Among the hematopoietic neoplasms, lymphoma and/or lymphosarcoma were the most frequent neoplasms in surveys of neoplastic lesions in cattle slaughtered in slaughterhouses [4,25]. Lymphoma constitutes one of the main bovine tumors, and it is associated with relevant economic losses in the production chain of this species [10]. Lymphoma was also the most frequently found neoplasm in cattle slaughtered in Canada [25] and Brazil [4]. Lymphoma is an enzootic neoplastic disease of lymphocytes that affects adult animals of various ages, but predominantly between 4 and 8 years [4,26]. Although lymphoma is more commonly diagnosed in dairy cattle, this tumor also causes losses in the beef industry, being responsible for the condemnation of organs or carcasses [27].

Ocular SCC is the most frequent neoplasm in cattle worldwide, being responsible for large economic losses due to the reduction in reproductive life or the condemnation of carcasses in slaughterhouses [28]. With 13.15% (172/1308) of the neoplasms extracted from the studies reviewed, the special senses (i.e., eyes and ears) were the fourth most affected organ system. According to this review, the most frequent neoplasm was ocular SSC. In a study on neoplasms carried out in Egypt [16], ocular SSC was also the most frequent [28,29].

According to this review, the other organ systems were less affected by neoplasia in cattle. Among these organ systems, the female and male genital systems account for 7.00% (92/1308) of the observed neoplasms. According to Lucena et al. [8], the female and male genital systems were also the sixth organ system most affected by neoplasms. In this review, the most frequent neoplasms were SSC, papilloma/fibropapilloma, and adenocarcinoma. These findings coincide with the results of a study conducted in Germany [30]. The US, with 4.97% (65/1308) of the observed neoplasms, showed transitional cell carcinoma as the most frequent neoplasm. These results corroborate a retrospective study that investigated 586 neoplasms in cattle, where transitional cell carcinoma and renal carcinoma were the most frequent [8].

Endocrine glands were responsible for 3.98% (52/1308) of the neoplasms reported in the reviewed studies, with pheochromocytoma being the most frequent neoplasm. Pheochromocytoma was the third most common neoplasm in samples of slaughtered cattle [4]. The world literature points to a low incidence of adrenal gland tumors in slaughtered cattle, and incidence may vary according to the geographic region, and even epigenetic factors may be involved [31].

The nervous system, with 2.90% (38/1308), had the most frequent neoplasm, the shwannoma, and the same was reported by [4,8,21]. The respiratory system, with 2.40% (31/1308), had lung adenocarcinoma as the most observed neoplasm; this neoplasm was also observed in samples of slaughtered cattle [8,21]. Liver and biliary systems were reported to be affected in 1.84% (24/1308) of the neoplasms presented in the reviewed studies, with hepatic adenocarcinoma being the most observed. Hepatocellular carcinoma was also reported in a study with slaughterhouse samples [32]. The other systems, such as bones and joints, muscles and tendons, the pancreas, and the cardiovascular system, represented less than 0.50% of the neoplasms contained in the reviewed studies. Studies referring to neoplasms in these systems are of low frequency or considered rare [2,4,8,32].

### 4.2. Goats

The goat species had the second highest number of neoplasms according to the studies reviewed, with 10.5% (197/1873), but no studies were found in slaughterhouses. The most affected organ system was the integumentary system with 48.22% (95/197), and the most frequent neoplasms were SCC with 40% (38/95) and melanoma with 25.26% (24/95). In the literature, existing studies show that the frequency of SCC in goats changes according to the region studied [5]. Data on the occurrence of SCC in goat farms in northern Brazil revealed a frequency of 3.08%, however, the total herd was 747 goats [33]. In the United States of America (USA), in a study carried out in a pathological anatomy laboratory, when examining 100 neoplasms in goats, 10% (10/100) were SCC [34]. In Saudi Arabia, in a study on cutaneous neoplasms in goats, the frequency of SCC was 15%; however, the total number of animals was 15 [35]. Skin and adnexal neoplasms are common in animals living in tropical countries, in particular SCC, due to chronic exposure of animals to ultraviolet radiation associated with fair skin [5].

Studies reporting cutaneous melanoma in goats are few, and, when available, they have a low number of diagnoses. Although melanocytic cutaneous neoplasms in small ruminants are rare, it is suggested that Angora is susceptible to melanoma [35]. Of the studies reviewed, most were case reports and/or retrospective studies associated with other species [13,34,36,37]. Another aspect that draws attention is that most melanocytic lesions and/or neoplasms can metastasize to other organs such as the liver, lung, kidney, sternum, and mandible [38,39].

The male and female genital systems were the second most affected organ systems in goats, with 13.71% (27/197) of the neoplasms identified in the reviewed studies. The most frequent neoplasms were SCC and adenocarcinomas located in the uterus and cervix, with 29.63% (8/27) each. According to the present review, SCC was described on the vulva of goats with depigmented skin in Brazil [33] and in Saudi Arabia [35]. In a USA study, among female and male genital system neoplasms, adenocarcinomas represented 11.11% (3/27) of diagnosed neoplasms [34]. Other neoplasms that affected the female and male genital systems with low frequency were metastatic seminoma in a Parda Alpina goat [40] and thecoma in an 8-year-old non-breed goat [41,42]. Female and male genital system neoplasms in small ruminants are rarely reported due to the early slaughter of these animals [43].

With 12.20% (24/197) of the reported cases, the hematopoietic system was the third organ system with more diagnoses of neoplasms, with lymphoma (15/197, 7.61%) and thymoma (9/15, 60%) being the most frequent neoplasms. Lymphomas and thymona in this species were reported in studies in the USA [34]. Pakistan and Brazil [13,43,44] reported isolated cases of lymphoma in goats [45].

The muscles and tendons, endocrine glands, nervous system, and liver and biliary system accounted for 2.03% (4/197) of the neoplasms raised in each. Rhabdomyosarcoma was the most frequent neoplasm in muscles and tendons, with 75% (3/4). However, this neoplasm is considered less common or rare, and the first description was made by Hogendoorn et al. [46]. According to the studies analyzed, three cases of rhabdomyosarcoma were found in goats [13,34]. In endocrine glands, pheochromocytoma was the most frequent neoplasm with 75% (3/4) of diagnoses, being a neoplasm considered uncommon in this species and more common in cattle and dogs [47]. The most frequent neoplasm of the nervous system was lymphoma 50% (2/4) [48].

Lymphoma was the most frequent neoplasm in goat liver and biliary tissues, with 1.52% (3/197). Hepatic lymphoma usually arises from metastases of multicentric lymphomas [49]. In the reviewed studies on neoplasms, pancreatic neoplasms were not observed, whereas in bones and joints, only one osteosarcoma was observed. Bone neoplasms such as osteosarcoma in goats are rare, reflecting the few reports available in the literature [50].

### 4.3. Sheep

Sheep were the third most affected species by neoplasms with 10.46% (196/1873), according to the studies consulted for this review. However, as with goats, no studies were found in slaughterhouses. The integumentary system was the most affected system and presented 59.19% (116/196) of the neoplasms observed in the studies included in the present review. Squamous cell carcinoma is the most frequent neoplasm, with 57.65% (113/196) of the integumentary system. Squamous cell carcinoma is relatively uncommon in sheep [5]. The low occurrence of this type of tumor in sheep is probably due to the fact that these animals are slaughtered before they reach middle age, which reduces their probability of developing neoplasms [51]. Squamous cell carcinoma has been reported in different regions of the world, such as Saudi Arabia [35], Uruguay [52], Brazil [7,13], Egypt [16], and Argentina [53].

With 26.02% (51/196), the alimentary system was the second most affected system by neoplasms in sheep. However, for this system, the only neoplasm observed was intestinal adenocarcinoma, according to the reviewed studies. The high occurrence of this type of neoplasm in sheep in New Zealand and Australia may be associated with the consumption of *P. aquilinum* [54,55]. The cases of intestinal adenocarcinoma in sheep included in this review were all described in New Zealand [11], with the exception of one incidental case diagnosed in Spain [56].

The respiratory system was the third most affected organ system, with 5.10% (10/196) of the latent neoplasms in the reviewed articles. The lung was the organ most affected by neoplasms, with lung adenocarcinoma being the most frequent neoplasm, accounting for 90% (9/10) of neoplasms observed in the sheep respiratory system. Ovine pulmonary adenocarcinoma is associated with a contagious disease, common in sheep but rarely affecting goats, that is caused by a beta-retrovirus known as Jaagsiekte sheep retrovirus (JSRV) [57,58]. Ovine pumonary adenocarcinoma is reported worldwide with the exception of Oceania [59].

According to the reviewed studies, the special senses had the fourth highest number of neoplasms at 4.08% (8/196), with SCC being the most common neoplasm at 75% (6/8). SCC is one of the most commonly diagnosed cutaneous neoplasms in sheep, although the special senses are not the most affected anatomical site [13]. Studies of risk factors for SCC in the semi-arid region of Brazil, by Carvalho et al. [13] and Macêdo et al. [60], suggested that in sheep, the skin of the head is the anatomical site most affected by SCC. However, SCC can affect other parts of the body, preferably hairless and depigmented areas, such as the eyelids, ears, snout, vulva, and perineum [61].

The most frequent tumors were those of the female and male genital systems and the liver and biliary systems, with 75% (3/4) each, respectively. The most frequent neoplasms were uterine adenocarcinomas and cholangiocarcinomas, with 1.02% (2/3) each, respectively. Uterine adenocarcinoma in ruminants is considered uncommon, being more commonly diagnosed in cattle and goats and rarely in sheep [30]. Cholangiocarcinoma in sheep is an incidental neoplasm, and it is rarely diagnosed [62].

The hematopoietic and nervous systems were affected by 1.02% (2/196) of the malignancies observed in the reviewed studies, respectively. In the hematopoietic system, 100% of the neoplasms identified in the reviewed studies were lymphomas. In sheep, lymphomas can be of idiopathic origin or due to infection with the bovine leukemia virus [63]. In the nervous system, the neoplasms found were an adenocarcinoma and a medulloblastoma [64]. Adenocarcinoma likely arose from metastasis from another tissue or organ [65]. The cardiovascular system, with 0.51% (1/196), was affected only by one hemangioma, which is rarely reported for this species [10].

### 4.4. Pigs

Of the studies included in the present review, 172 neoplasms were observed in swine, representing 9.18% (172/1873) of the neoplasms observed in the studies included in the present review. In this species, the most affected organ system was the integumentary system with 39.50% (68/172), with melanoma and melanocytoma being the most frequent neoplasms with 19.77% (34/172) and 16.86% (29/172), respectively. According to Bundza and Felmate [66], melanocytic lesions are found in pigs slaughtered for consumption, and it is necessary to distinguish whether they are melanosis or melanocytic neoplasms and whether they are benign or malignant in order to make an adequate sanitary decision about the carcass and the respective viscera. Data on the occurrence of these lesions in production animals are rare [7]. In a study carried out in Portuguese slaughterhouses, frequencies of 4.81% were found for melanocytomas and 21.15% for melanomas [67]. In Brazil, Brum et al. [68], when studying 37 swine neoplasms, found a frequency of 21.70% for melanomas. Studies available in the literature suggest that melanomas are congenital or that they appear in the first weeks of the pig’s life [51,68]. Melanocytic lesions in pigs spontaneously regress without relapse in approximately 85–90% of cases; it is suggested that this regression mechanism is a cell-mediated immune response [5,69]. However, some animals may end up dying as the disease progresses [70].

The hematopoietic system was the second most affected with 16.90% (29/172) of neoplasms in pigs, and the most frequent neoplasm with 86.20% (25/29) was lymphoma/lymphosarcoma [71,72].

The third system with the highest number of neoplasms was liver and biliary, which represented 16.30% (28/172) of cases according to the articles included in this review [73]. Hepatocellular carcinoma was the most common neoplasm, accounting for 85.71% (24/28) of all liver and biliary system neoplasms. Primary pig liver neoplasms accounted for 8.20% in a study carried out with samples from slaughterhouses [68]. There are descriptions in which hepatocellular carcinoma presents as single tumors or diffuse masses through the liver parenchyma; this second presentation predominates in the descriptions in pigs [74,75].

The urinary system was the fourth most affected system, with 9.88% of neoplasms in swine. The most frequent neoplasm was nephroblastoma, accounting for 94.12% (16/172) of neoplasms within the USA. In a study carried out in Brazil, the frequency of nephroblastoma was 29.70% [68]. Nephroblastoma is a common neoplasm in swine, found most often at slaughter, and it is the main primary renal neoplasm of the species; however, it varies depending on the region [51]. Reports found in the literature indicate that in the United States the incidence is estimated to be 20 nephroblastomas per 100,000 pigs, but in Russia it is 0.35 tumors per 100,000 pigs [74].

The male and female genital systems were affected by 8.72% (15/172) of neoplasms in swine. Leiomyoma was the most frequent neoplasm, accounting for 80.00% (12/15) of female and male genital system neoplasms. The prevalence of uterine leiomyomas in pigs is very low and may vary according to the age of the population studied [11]. The potbellied pig seems to have a high incidence of uterine leiomyomas due to the longevity of the species; the opposite occurs with the pig that goes to slaughter very young [75].

The alimentary system was the sixth most affected system, with 5.20% (9/172) of the neoplasms observed in swine. Lymphoma/lymphosarcoma was the most frequent neoplasm, with 44.44% (4/9) of AS. However, one study observed a frequency of 29.70% of lymphoma/lymphosarcoma in pig samples from slaughterhouses [68]. According to Charles [76], lymphoma/lymphosarcoma is commonly found in pigs during slaughter. However, a study carried out at a slaughterhouse in the province of Ontario in Canada between 1996 and 2003 indicates that 0.02 to 0.10 carcasses per 10,000 pigs that were slaughtered were condemned due to the fact that they were found to have lymphosarcoma [77].

According to the studies reviewed, the endocrine glands were affected by 1.16% (2/172) of the neoplasms, with pheochromocytoma being the only one observed. Pheochromocytomas are the most common neoplasms arising in the adrenal medulla of animals, and they develop most frequently in cattle and dogs, rarely in other domestic animals, and are extremely rare in swine [78,79]. In a survey carried out in slaughterhouses of 3.7 million pigs in the UK, only a 2.5-year-old female was diagnosed with pheochromocytoma, which shows a low incidence in pigs probably due to the fact that many animals are slaughtered before reaching maturity [80].

The nervous system, respiratory system, cardiovascular system, and bones and joints were affected by 0.58% (1/172), according to the studies included in this review. In the nervous system, the neoplasm observed was neuroblastoma, which belongs to the group called peripheral nerve sheath tumors [5]. Neurofibromas are composed of a mixture of Schwann cells, perineural cells, and fibroblasts [81]. In domestic animals, neurofibromas have been described predominantly in dogs and cattle and rarely in other species [82]. Lymphomas/lymphosarcomas have been observed only in the respiratory and cardiovascular systems; however, the appearance of this type of neoplasm is associated with metastasis [15]. In slaughter pigs, the incidence of lymphoma/lymphosarcoma varies according to aspects such as the region of study and the age of the animals [73]. A granulocytic sarcoma, a neoplasm rarely reported in the literature, has been described in bones and joints [68], but also in the liver, kidneys, and mesenteric lymph nodes [83].

## 5. Conclusions

Based on the results of this review, we can conclude that the bovine species was the most affected by neoplasia and also the most studied in relation to small ruminants and swine. In all species, the most affected organ system was the integumentary, and the most frequent neoplasms were the SCC in cattle, goats, and sheep, and melanoma in swine. It was also observed that there were a few studies carried out in slaughterhouses, and those that were found referred to cattle and swine; none of them mentioned goats and sheep. No studies were found that measured the economic losses associated with carcass condemnation of the species studied. In view of the above, it is necessary to carry out studies in farms and slaughterhouses that provide more information, such as the total number of animals and the origin of the samples. Broad and detailed studies will provide knowledge about the impact of neoplasms on production and carcass condemnation in ruminants and swine and the respective risk factors.

## Figures and Tables

**Figure 1 vetsci-10-00163-f001:**
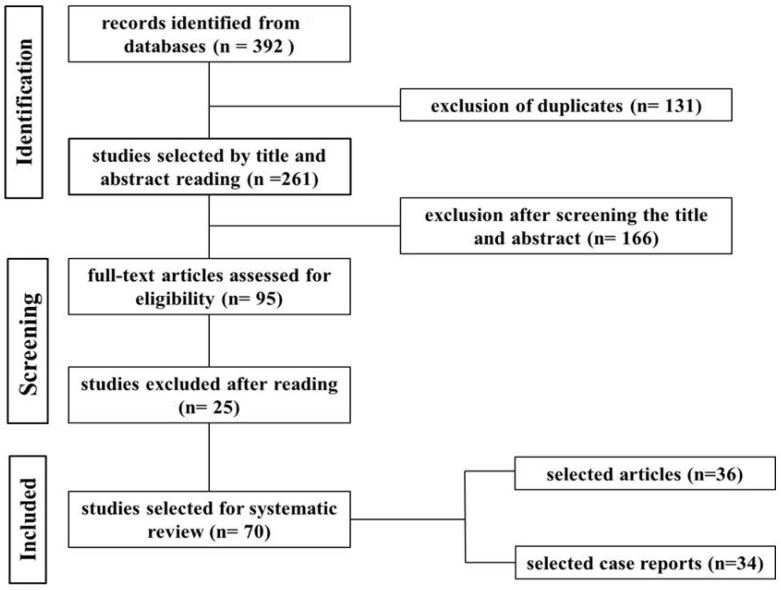
Flowchart of the search, selection, and inclusion of studies in the systematic review based on the PRISMA 2020 guidelines.

**Table 1 vetsci-10-00163-t001:** Number of studies, according to the country where they were carried out and the species affected by neoplasms.

Species and Number of Articles					
Countries	Cattle	Goats	Sheep	Pigs	Total
Algeria			1		1
Argentina		1	1		2
Brazil	12	6	3	2	23
Canada				2	2
Croatia	1				1
Czech Republic	1				1
Denmark	1				1
Egypt	2	1	2		5
Ethiopia	1				1
Germany	1			1	2
India	1	1			2
Iran			1		1
Italy		1		2	3
Japan	3			3	6
Pakistan		1	1		2
Portugal				1	1
Saudi Arabia		1	1		2
Spain		1	2	5	8
Turkey		1	2		3
United States	2	4	1	4	11
Uruguay			1		1
Total (%)	25 (31.6%)	18 (22.8%)	16 (20.3%)	20 (25.3%)	79 (100%)

**Table 2 vetsci-10-00163-t002:** Origin of the cases of neoplasms included in the studies according to the species infected.

Origin of Samples	Species				
	Cattle	Goats	Sheep	Pigs	Total
Farm	3	4	5	0	12
Slaughterhouse	5	2	3	8	18
Not informed	17	12	8	12	49
Total (%)	25 (31.6%)	18 (22.8%)	16 (20.3%)	20 (25.3%)	79 (100%)

**Table 3 vetsci-10-00163-t003:** Classification in epithelial, mesenchymal, and embryonic neoplasms.

Classification/n° of Cases			
Species	Epithelial	Mesenchymal	Embryonic
Cattle	930	373	5
Goats	94	102	1
Sheep	187	8	1
Pigs	37	119	16
Total (%)	1248 (66.6%)	602 (32.1%)	23 (1.23%)

**Table 4 vetsci-10-00163-t004:** Distribution of neoplasms in cattle according to the organ system.

Type of Neoplasm	Organ System														
	AS	IS	FMGS	US	MT	EG	HS	NS	SS	RS	LBS	CS	BJ	P	Total
	n°	n°	n°	n°	n°	n°	n°	n°	n°	n°	n°	n°	n°	n°	n°/(%)
Squamous cell carcinoma	183	97	38	1					169	3					491 (37.5%)
Papilloma/fibropapilloma	29	216	15	1					2	1					264 (20.2%)
Adenocarcinoma	9	1	13			8				9				1	41 (3.1%)
Mesothelioma	7														7 (0.53%)
Fibrosarcoma	5	4	2	1						2					14 (1.0%)
Fibroma	3	15	2	1											21 (1.6%)
Lymphoma/lymphosarcoma	2	1					198								201 (15.4%)
Epulide	2														2 (0.15%)
Lipoma	1				1										2 (0.15%)
Gastrointestinal stromal tumor	1														1 (0.08%)
Leiomyosarcoma	1		2	1											4 (0.30%)
Undifferentiated carcinoma	1								1						2 (0.15%)
Ameloblastic fibro-odotoma	1														1 (0.08%)
Adenoma	1		1			2				1	2				7 (0.53%)
Melanoma		30													30 (2.29%)
Myxoma/fibromyxoma		3								1					4 (0.30%
Melanocytoma		2													2 (0.15%)
Myxoid liposarcoma		1													1 (0.08%
Histiocytic sarcoma		1													1 (0.08%)
Hemangioma		1	2	9											12 (0.90%)
Hemangiosarcoma		1		12						3	2	2			23 (1.76%)
Granulosa cell tumor			8												8 (0.60%)
Leiomyoma/fibroleyomioma			4		2										6 (0.46%)
Mamary carcinoma			2												2 (0.15%)
Teratoma			2												2 (0.15%)
Luteoma			1												1 (0.08%)
Transitional cell carcinoma				25											25 (1.90%)
Renal cell carcinoma				15											15 (1.15%)
Pheochromocytoma						26									26 (2.0%)
Adrenal cortex carcinoma						13									13 (1.0%)
Chromophobic adenoma						1									1 (0.08%)
Thyroid carcinoma						1									1 (0.08%)
Paraganglioma						1									1 (0.08%)
Schwannoma								23							23 (1.80%)
Neurofibroma								10							10 (0.80%)
Ependymoma								2							2 (0.15%)
Choroid plexus carcinoma								1							1 (0.08%)
Neuroblastoma								1							1 (0.08%)
Meningioma								1							1 (0.08%)
Carcinomas										9				1	10 (0.80%)
Pulmonary blastoma										1					1 (0.08%)
Neuroendocrine tumor										1					1 (0.08%)
Hepatoma											1				1 (0.08%)
Hepatocellular carcinoma											15				15 (1.15%)
Cholangiocarcinoma											4				4 (0.30%)
Chondroma													2		2 (0.15%)
Osteosarcoma													3		3 (0.23%)
Insulinoma														1	1 (0.08%)
Total	246	376	92	65	3	52	198	38	172	31	24	2	6	3	1308 (100%) (0.0%) 100
%	18.8%	28.7%	7.0%	5.0%	0.2%	4.0%	15.1%	3.0%	13.1%	2.4%	2.0%	0.1%	0.4%	0.2%	100%

AS (alimentary system); IS (integumentary system); FMGS (female and male genital systems); US (urinary system); MT (muscles and tendons); EG (endocrine glands); HS (hematopoietic system); NS (nervous system); SS (special senses); RS (respiratory system); LBS (liver and biliary system); CS (cardiovascular system); BJ (bones and joints); P (pancreas).

**Table 5 vetsci-10-00163-t005:** Distribution of neoplasms in goats according to the organ system.

Type of Neoplasm	Organ System														
	AS	IS	FMGS	US	MT	EG	HS	NS	SS	RS	LBS	CS	BJ	P	Total
	n°	n°	n°	n°	n°	n°	n°	n°	n°	n°	n°	n°	n°	n°	n°/(%)
Squamous cell carcinoma		38	8						15						61 (31.1%)
Papilloma		2													2 (1.0%)
Adenocarcinoma	5		8							5					18 (9.1%)
Fibrosarcoma		2	3												5 (2.6%)
Fibroma		2													2 (1.0%)
Lymphoma	1						15	2		3	3				25 (12.7%)
Thymoma							9								9 (4.6%)
Melanoma		24	3												27 (13.7%)
Myxoma		1													1 (0.5%)
Liposarcoma		2													2 (1.0%)
Hemangioma		6													6 (3.0%)
Hemangiossarcoma		7													7 (3.6%)
Leiomyoma			3												3 (1.5%)
Pheochromocytoma						3									3 (1.5%)
Choroid plexus carcinoma								1							1 (0.5%)
Cholangiocarcinoma											1				1 (0.5%)
Chondrosarcoma					1										1 (0.5%)
Osteosarcoma													1		1 (0.5%)
Odontogenic tumor	1														1 (0.5%)
Gingival sarcoma	2														2 (1.0%)
Signet ring carcinoma	1														1 (0.5%)
Sebaceous carcinoma		1													1 (0.5%)
Mast cell tumor		5													5 (2.6%)
Histiocytoma		2													2 (1.0%)
Apocrine sweat gland adenoma		1													1 (0.5%)
Tecoma			1												1 (0.5%)
Seminoma			1												1 (0.5%)
Sebaceous epithelioma		2													2 (1.0%)
Rhabdomyosarcoma					3										3 (1.5%)
Thyroid carcinoma						1									1 (0.5%)
Peripheral nerve sheath tumor								1							1 (0.5%)
Total n°	10	95	27		4	4	24	4	16	8	4		1		197 (100%)
%	5.1%	48.2%	13.8%		2.0%	2.0 %	12.2%	2.0%	8.1%	4.1%	2.0%		0.5%		100%

AS (alimentary system); IS (integumentary system); FMGS (female and male genital system); US (urinary system); MT (muscles and tendons); EG (endocrine glands); HS (hematopoietic system); NS (nervous system); SS (special senses); RS (respiratory system); LBS (liver and biliary system); CS (cardiovascular system); BJ (bones and joints); P (pancreas).

**Table 6 vetsci-10-00163-t006:** Distribution of neoplasms in sheep according to the organ system.

Type of Neoplasm	Organ System														
	AS	IS	FMGS	US	MT	EG	HS	NS	SS	RS	LBS	CS	BJ	P	Total
	n°	n°	n°	n°	n°	n°	n°	n°	n°	n°	n°	n°	n°	n°	n°/(%)
Squamous cell carcinoma		113							6						119 (60.8%)
Papilloma		1													1 (0.5%)
Adenocarcinoma	51		2					1		9					63 (32.1%)
Mamary fibroadenoma			1												1 (0.5%)
Lymphoma							2								2 (1.0%)
Melanoma		1							2	1					4 (2.1%)
Myxoma		1													1(0.5%)
Hemangioma												1			1(0.5%)
Cholangiocarcinoma											2				2 (1.0%)
Hepatic carcinoma											1				1 (0.5%)
Medulloblastoma								1							1(0.5%)
Total n°	51	116	3				2	2	8	10	3	1			196 (100.0%)
%	26.1%	59.2%	1.5%				1.0%	1.0%	4.1%	5.1%	1.5%	0.5%			100.0%

AS (alimentary system); IS (integumentary system); MFGS (female and male genital system); US (urinary system); MT (muscles and tendons); EG (endocrine glands); HS (hematopoietic system); NS (nervous system); SS (special senses); RS (respiratory system); LBS (liver and biliary system); CS (cardiovascular system); BJ (bones and joints); P (pancreas).

**Table 7 vetsci-10-00163-t007:** Distribution of neoplasms in swine according to the organ system.

Type of Neoplasm	Organ System														
	AS	IS	FMGS	US	MT	EG	HS	NS	SS	RS	LBS	CS	BJ	P	Total
	n°	n°	n°	n°	n°	n°	n°	n°	n°	n°	n°	n°	n°	n°	n° (%)
Squamous cell carcinoma	2	2													4 (2.3%)
Fibroma	1														1 (0.6%)
Papilloma		2													2 (1.1%)
Adenocarcinoma	1		1												2 (1.1%)
Lymphoma/lymphosarcoma	4			1			25			1	1	1			33 (19.2%)
Melanoma		34													34 (19.8%)
Melanocytoma		29													29 (16.9%)
Mast cell tumor		1													1 (0.6%)
Pheochromocytoma						2									2 (1.1%)
B lymphoblastic leukemia							1								1 (0.6%)
Fibrous histiocytoma							3								3 (1.7%)
Neurofibroma								1							1 (0.6%)
Nefhroblastoma				16											16 (9.3%)
Cholangiocarcinoma											1				1 (0.6%)
Hepatocellular adenoma											2				2 (1.1%)
Carcinoma hepatocellular											24				24 (14.0%)
Granulocytic sarcoma													1		1 (0.6%)
Ganglioneuroma	1														1 (0.6%)
Leiomyoma			12												12 (7.0%)
Leiomyosarcoma			1												1 (0.6%)
Undifferentiated sarcoma			1												1 (0.6%)
Total n°	9	68	15	17		2	29	1		1	28	1	1		172 (100%)
	5.2%	39.5%	8.7%	9.9%		1.1%	16.9%	0.6%		0.6%	16.3%	0.6%	0.6%		100%

AS (alimentary system); IS (integumentary system); FMGS (female and male genital system); US (urinary system); MT (muscles and tendons); EG (endocrine glands); HS (hematopoietic system); NS (nervous system); SS (special senses); RS (respiratory system); LBS (liver and biliary system); CS (cardiovascular system); BJ (bones and joints); P (pancreas).

## Data Availability

Data are available from the corresponding author upon reasonable request.

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
