# Peer review of "Neoplasms in Domestic Ruminants and Swine: A Systematic Literature Review"

_vetsci, 2023, doi:10.3390/vetsci10020163_

Round 1
Reviewer 1 Report
Neoplasms in ruminants and swine can lead to economic losses to agricultural enterprises due causing poor animal performance and even condemnation of carcasses. The present review clearly illustrates that little is known about the incidence of neoplasms in cattle, goats, sheep and pigs, and that more intense study is likely warranted.
I have no problem with the material included or the overall organization. My comments will be editorial in nature.
The biggest thing is that the authors should never use acronymes for the different organ systems anywhere in the manuscript text. These are very confusing and the organ system names should be completely typed out throughout the text.
Line 44 - "increase"
Lines 48-49 - "comparison with" what?
Line 80 "case"
Line 121 "added"
Line 123&124 - "of studies"
Line 127 "are embryonic"
Line 128 - "A summary of the distribution"
Line 135 - "neoplasm"
Line 142 - "organ"
Line 152 - "with 10.52%"
Lines 157-166 - in all cases it should be "integumentary"
Like 161 - "detailed" rather than 'distributed"?
Line 254 - "increase" instead of "determine"?
Line 262 - "diagnosed"
Line 268 - remove the comma
Line 271 - "previous study"\Line 284 - "result was previously"
Line 301 - move comma to behind "BPV-4"
Line 303 - "The hematopoietic"
Line 313 - "in the beef"
Line 321 - "and environmental"
Line 324 - "and" instead of "it is"?
Line 329 - "showed"
Line 330 - remove comma
Line 336 - "and incidence" instead of "because they"?
Line 341 - "Cattle by"?
Line 342 - "with" rather than "being"?
Line 355 - "observed the frequency"
Line 356 - "America"
Line 369 - "systems were"
Line 371 - "uterus"
Line 373 - "and in Saudi" and "In a USA"
Line 376 - "breeder"?
Line 382 - "Pakistan and Brazil"
Line 401 - "Sheep"
Line 419 - "New Zealand by"?
Line 425 - "that is caused"
Line 431 - "Studies of risk factors"
Line 434 - "other parts of the body"
Line447 - change "to" to "from"?
Line 458 - "they are" in both palces
Line 463 - add comma after neoplasms
Line 464 - remove "melanoma"
Line 475 - "condemnations" rather than "convictions"?
Line 482 - "tumors or as diffuse"
Line 487 - "The urinary system"
Line 489 - "plasms within the US"
Line 490 - "e"? and replace ", it" with "and"
Line 495 - "the male " and "systems were"
Line 510 - move "being" to behind "pheochromocytoma"
Lines 522-524 - sentence does not make sence
Line 526 - "BJ, but this"
Author Response
Dear Editor, on behalf of all the authors, thank you for your editorial work. We also thank very much the comments and suggestions by Reviewer #1 and Reviewer #2, which have allowed us to improve our manuscript. Responses to all the points raised by the Reviewers can be find below.
With kind regards, Jackson Vasconcelos.

Reviewer 2 Report
Comment and suggestions are in the attached word file.

Author Response
Prezado Editor, em nome de todos os autores, obrigado por seu trabalho editorial. Agradecemos também os comentários e sugestões do Revisor nº 1 e do Revisor nº 2, que nos permitiram aprimorar nosso manuscrito. As respostas a todos os pontos levantados pelos revisores podem ser encontradas abaixo. Com os melhores cumprimentos, Jackson Vasconcelos.
